# Peptide-Templated Gold Clusters as Enzyme-Like Catalyst Boost Intracellular Oxidative Pressure and Induce Tumor-Specific Cell Apoptosis

**DOI:** 10.3390/nano8121040

**Published:** 2018-12-12

**Authors:** Ya Zhang, Xiangchun Zhang, Qing Yuan, Wenchao Niu, Chunyu Zhang, Jiaojiao Li, Zhesheng He, Yuhua Tang, Xiaojun Ren, Zhichao Zhang, Pengju Cai, Liang Gao, Xueyun Gao

**Affiliations:** Department of Chemistry and Chemical Engineering, Beijing University of Technology, No.100, Pingleyuan, Chaoyang District, Beijing 100124, China; zhangya1@ihep.ac.cn (Y.Z.); zhangxc@ihep.ac.cn (X.Z.); yuanqing@bjut.edu.cn (Q.Y.); niuwc@ihep.ac.cn (W.N.); zhangcy@ihep.ac.cn (C.Z.); lijiaojiao@ihep.ac.cn (J.L.); hezs@ihep.ac.cn (Z.H.); tangyh@ihep.ac.cn (Y.T.); xjren@bjut.edu.cn (X.R.); zhangzc@ihep.ac.cn (Z.Z.); caipj@ihep.ac.cn (P.C.)

**Keywords:** gold clusters, biocatalysis, reactive oxygen species, oxidative pressure-type tumor

## Abstract

Anticancer metallodrugs that aim to physiological characters unique to tumor microenvironment are expected to combat drug tolerance and side-effects. Recently, owing to the fact that reactive oxygen species’ is closely related to the development of tumors, people are committed to developing metallodrugs with the capacity of improving the level of reactive oxygen species level toinduce oxidative stress in cancer cells. Herein, we demonstrated that peptide templated gold clusters with atomic precision preferably catalyze the transformation of hydrogen peroxide into superoxide anion in oxidative pressure-type tumor cells. Firstly, we successfully constructed gold clusters by rationally designing peptide sequences which targets integrin α_ν_β_3_ overexpressed on glioblastoma cells. The superoxide anion, radical derived from hydrogen peroxide and catalyzed by gold clusters, was confirmed in vitro under pseudo-physiological conditions. Then, kinetic parameters were evaluated to verify the catalytic properties of gold clusters. Furthermore, these peptide decorated clusters can serve as special enzyme-like catalyst to convert endogenous hydrogen peroxide into superoxide anion, elevated intracellular reactive oxygen species levels, lower mitochondrial membrane potential, damage biomacromolecules, and trigger tumor cell apoptosis consequently.

## 1. Introduction

Reactive oxygen species (ROS) induced oxidative stress has shown profound impact on the fate of tumor, including occurrence, development and extinction [1,2,3,4,5]. It is well estimated that in the tumor microenvironment, the hydrogen peroxide (H_2_O_2_) level (50–100 µM) far exceeds those of normal tissues [6,7]. In view of this point, multifarious antitumor agents, including various nanomaterials, with the aim of targeting the overexpressed H_2_O_2_, have been developed in recent years. Notably, as pioneering reports, iron oxide nanoparticles were rendered as peroxidase-mimic enzymes to initiate fenton reaction under the mild acidic tumor environment to trigger cellular apoptosis [8,9]. In detail, these nanoparticles decomposed high-levelled H_2_O_2_ within tumor cells into ROS, such as hydroxyl radical (∙OH) with much higher activity.

Due to their excellent biocompatibility and biosafety, gold-based nanomaterials including nanoclusters, nanospheres, nanorods, nanostars and nanocages, are considered as potential candidates for wide bioapplications such as bioimaging, bioanalysis and biomedicine [10,11]. Among them, gold clusters (AuCs) represent a stable transition substance between large nanoparticles and atoms. Thanks to their ultrasmall size and precise structure, AuCs possesses featured physical and chemical properties including electromagnetic behaviour, fluorescence and catalytic activities [12,13]. Currently, it is found that various chemical reactions can be catalyzed by AuCs with high selectivity and efficiency [14]. However, most of these reactions are carried out in relatively harsh conditions compared to tumor microenvironment. To our best knowledge, AuCs-initiated biocatalysis under mild physiological condition for anticancer treatment has scarcely been reported.

In this article, we explore AuCs modified by a small peptide targeting integrin α_ν_β_3_ to selectively boost the tumor-specific oxidative pressure with higher efficiency. The gold clusters have intrinsic fluorescence characteristics and precise molecular composition, as well as catalytic properties. As shown in Scheme 1, through selective recognition between the peptide and integrin, peptide-modified AuCs enter the cell and then catalyze H_2_O_2_ to produce superoxide anion radicals (O_2_^−●^). It results in the remarkable increase of the ROS level in tumor cells, stimulating mitochondrial apoptosis pathway to release caspases and induce apoptosis.

## 2. Materials and Methods

### 2.1. Materials and Reagents

The peptide H_2_N–CGPDGRDGRDGPDGR–COOH was synthesized by a solid-phase method with purity at around 98% (China Peptides Co. Ltd., Hangzhou, China). Hydrogen tetrachloroaurate (III) (HAuCl_4_·4H_2_O), sodium hydroxide (NaOH), hydrogen peroxide (H_2_O_2_), nitric acid (HNO_3_) and hydrochloric acid (HCl) were obtained from Sinopharm Chemical Reagent Co. Ltd. (Shanghai, China). Human malignant glioblastoma cell (U87-MG) and Henrietta Lacks strain of cancer cells (HeLa) were obtained from the Cancer Institute and Hospital, Chinese Academy of Medical Sciences (Beijing, China). Phosphate buffer solution (PBS), cell culture medium DMEM, and MEM were purchased from Hyclone (South Logan, UT, USA). Fetal bovine serum (FBS) and trypsin-EDTA were gained from Gibco (Grand Island, NY, USA). The ROS probe 5-(and-6)-chloromethyl-2′, 7′-dichlorodihydrofluorescein diacetate acetyl ester (CM–H_2_DCFDA) was from Molecular Probes (Eugene, OR, USA). The agent 5,5′,6,6′-tetrachloro-1,1′,3,3′-tetraethylbenzimidazolocarbocyanine iodide (JC-1), BCA Protein Assay Kit, and cell lysis buffer were purchased from Beyotime Institute of Biotechnology (Haimen, China). Apoptosis Detection Kit (Annexin V-FITC/PI) and cell counting kit-8 (CCK-8) were acquired from Dojindo Laboratories (Kumamoto, Japan). β-actin antibody, PARP antibody, anti-rabbit IgG-HRP and caspase-3 antibody were purchased by Cell Signaling Technology (Danvers, MA, USA). Ultrapure water (18 MΩ) was obtained from a Milli-Q synthesis (Millipore Corp, Burlington, MA, USA) system that was used throughout the experiment.

### 2.2. Preparation of Peptide-Au Clusters

The peptide (6 mg) was dissolved in 1 mL ultrapure water, and HAuCl_4_ (25 mM, 40 μL) was slowly added into the peptide solution at an ambient temperature. Five minutes later, 200 μL of 0.5 M NaOH was added to the above solution. The reaction was allowed to take place for 12 h at 55 °C. Superfilter tube (MWCO: 3 kDa) was employed to purify the sample in order to remove free peptide and ions.

### 2.3. Characterization of Peptide-Au Clusters

Ultraviolet-visible absorption spectra of AuCs were measured by a spectrophotometer (Shimadzu UV-1800, Kyoto, Japan). The fluorescence spectra of AuCs were acquired by a fluorescence spectrophotometer (Shimadzu RF-5301, Japan). The high-resolution transmission electron microscopy (HRTEM, JEM-2100, JEOL Ltd., Akishima, Japan) was employed to measure the size distribution of synthesized AuCs. Practically, a drop of AuCs aqueous solution was transferred and dried on ultrathin carbon-coated copper grids for HRTEM studies. Matrix-assisted laser desorption/ionization time of flight mass spectrometry (MALDI-TOF MS, UltrafleXtreme, Bruker, Bremen, Germany) was used to determine the molecular formula of synthesized AuCs in the positive ion linear mode.

### 2.4. Inductively Coupled Plasma Mass Spectrometry Analysis of Peptide-Au Clusters’ Concentration

The concentration of synthesized AuCs was acquired by inductively coupled plasma mass spectrometry (ICP-MS) analysis system (Thermo Elemental X7, Waltham, MA, USA). The thoroughly purified AuCs (20 μL) was predigested by themixture of HNO_3_ and HCl (volume ratio 3:1) overnight. Then, the mixture was evaporated to approximate 0.2 mL and diluted to 2500-fold by 2% HNO_3_, and 1% HCl. Bismuth (20 ppb) solution was monitored as an internal standard. The standard curve was acquired by measured a series of standard Au aqueous solutions (0.1, 0.5, 1, 5, 10, 50, 100 ng mL^−1^ in 2% HNO_3_ and 1% HCl dilute solution). The prepared samples were measured by ICP-MS system.

### 2.5. Superoxide Anion Detection

Nitroblue tetrazolium (NBT) can form formazan that insoluble in water when reacts with superoxide anion, which can be used to detect superoxide anion. Firstly, 300 μM H_2_O_2_, 3 mM NBT was mixed in equal volume with 100, 300, 500 μM AuCs to obtain a 300 μL of solution, respectively. The mixture reacted for 30 min under darkness before centrifuged at 7200 rpm for 10 min. Then, the supernatant was discarded and the precipitate was dissolved by DMSO (100 μL). The absorption of the product was recorded.

### 2.6. Optimal Catalytic Temperature and pH

To investigate the optimal catalytic pH condition, 100 μM H_2_O_2_, 5 μM AuCs and 50 μM Amplex UltraRed (AUR) was mixed at 37 °C under pH 3.0, 5.0, 7.0, 9.0 and 11.0 conditions, respectively. The mixture was allowed to react for 30 min at 37 °C under darkness. The fluorescence of the mixture was measured between 470 and 800 nm afterwards. The oxidation of AUR in the presence of H_2_O_2_ catalyzed by AuCs produced fluorescence signals with maximum emission peaks at 585 nm. After the ideal pH condition was determined, the similar reaction was initiated at the temperature of 25, 30, 35, 40, 45 °C to obtain the ideal catalytic temperature under 7.0 pH condition.

### 2.7. Steady-State Kinetic Studies

An AUR reagent was employed as substrate for the following enzyme activity assays. The volume of 40 μL H_2_O_2_ (25, 50, 100, 250, 750, 1000 μM) was used as a varying substrate. AUR solution (25 μM, 40 μL) was added into 20 μL of 5 μM clusters solution. The reaction proceeded in a 96-well plate for 30 min at 37 °C. Fluorescence of the mixture was measured at 585 nm on a microplate reader (SpectraMAX M2, Sunnyvale, CA, USA). The Michaelis-Menten constant and the maximum reaction rate were determined by the profiled Lineweaver-Burk plot.

### 2.8. Cell Viability Assay

5 × 10^3^ cells/well U87-MG and HeLa cells were inoculated on 96-well plates and cultured for 24 h. Gradient doses of free peptides and AuCs were administrated into wells and incubated with cells for 48 h. The peptide concentration is calculated by the ratio (7:5) of peptide to gold on the basis of the corresponding AuCs concentration. Then, cells were cocultured with fresh medium containing 10% (v/v) CCK-8 reagent and incubated for 30 min at 37 °C after washed three times with PBS. The microplate reader was employed to record the absorbance of each well at OD_450 nm_.

### 2.9. Cellular Location of Peptide-Au Clusters

U87-MG and HeLa cells were seeded on glass-bottom dishes and cultured in a cell culture incubator for 24 h. After that, 40 μM AuCs was added and incubated with cells for 24 h before they were slightly rinsed with PBS three times and cultured with 20 nM lysotracker for 30 min. Then cells were washed and observed by CLSM with an excitation wavelength at 405 nm and 560 nm, respectively. Cellular location of AuCs can be determined by the colocalization.

### 2.10. Analysis of Intracellular ROS Level

To detect intracellular ROS level, U87-MG cells firstly grew on the glass-bottom dishes for 24 h. Then, 40 µM AuCs was added into culture medium and incubated for a series of time (3, 6, 12 h). After, the cells were stained that 5 µM CM-H_2_DCFDA before by washed with PBS. Then, cells were rinsed with PBS and supplemented with fresh culture medium. The ROS level generated in cells is observed by the green fluorescence intensity with a 488 nm laser irradiation.

### 2.11. Apoptosis Evaluation

Annexin V-FITC/PI fluorescence dual staining was used to evaluate apoptosis/necrosis situation. 1 × 10^5^ cells/mL U87-MG cells were seeded in 6-well plates overnight before cultured with clusters at a series of Au concentration for 48 h. U87-MG cells were then collected by centrifuging at 1000 rpm for 5 min. Then, the cells were gently washed with PBS. The volume of 195 μL of binding buffer was added to the mixed cells. After that, the cells were incubated with 5 μL of Annexin V-FITC and 10 μL of PI solution and stained under darkness for 15 min. Apoptosis/necrosis cells were then detected by flow cytometer. Experimental results were further analyzed by flow cytometer software.

### 2.12. Mitochondrial Membrane Potential (ΔΨm) Assay

The decrease of cell membrane potential was monitored by the fluorescence transition of JC-1 aggregates to JC-1 monomers with red emission and green emission, which can be used as a detection index for early apoptosis. U87-MG cells were seeded on glass-bottom dishes and cultured for 24 h in a cell culture incubator. Then, AuCs (40 µM) was added to culture medium and cultured for a series of time (3, 6, 12 h). Cells were incubated with JC-1 for 20 min under darkness. Then, cells were washed twice with PBS. JC-1 monomers and JC-1 aggregates were observed at the green emission channel and the red emission channel when exited at 488 nm.

### 2.13. Western Blot Analysis of Apoptotic Protein Expression

U87-MG cells were incubated with AuCs containing Au concentration at 0, 20, 40, 60 μM for 48 h in 6-well plates. Then, the medium was discarded and cells were rinsed with PBS. Cells were lysed in RIPA buffer with 1% cocktail for 20 min at 4 °C. The supernatant was collected for 5 min at 12,000 g. Protein concentrations were measured by the BCA assay kit. Five-fold loading buffer was diluted to 1× loading buffer with protein sample. After boiling for 5 min, the equal amount of protein was added into 10% SDS-polyacrylamide gel for separation and then transferred onto PVDF membranes. Membranes were blocked with blocking solution at room temperature for 2 h and then incubated with the caspase-3, caspase-7, PARP antibodies at 4 °C overnight. Then, horseradish peroxidase-conjugated secondary antibody was used to incubate the PVDF membranes at room temperature for 1 h. The result of antibody-antigen reactions was detected with Amersham ECLTM Prime Western Blotting Detection Reagent (GE Healthcare, UK).

### 2.14. Single Cell Quantitative Analysis

The AuCs concentration enveloped by a single U87-MG cell was calculated by dividing the number of cells by the determined AuCs concentration in the massive cells. For example, single cell quantitative analysis in the condition of incubation HeLa cells with 40 μM AuCs for 24 h was performed through ICP-MS characterization. The formula is present.
$$c=\frac{\rho V}{M_{Au}n_{\mathrm{cell}}V_{\mathrm{cell}}}=\frac{{4.99\times 10}^{-6}\times {5.0\times 10}^{-3}}{197\times {5.0\times 10}^{5}{\times 7\times 10}^{-12}}=3.5\times 10^{-5}$$
c: average AuCs concentration in single cell (M)ρ: Au ion mass concentration determined from ICP-MS analysis (g L^−1^)V: analyzed solution volume (L)M_Au_: relative atomic mass of Au (g mol^−1^)n_cell_: U87-MG cells numberV_cell_: single U87-MG cell volume (L)


## 3. Results and Discussion

### 3.1. Characterization of Peptide Templated Gold Clusters

In this article, we explore AuCs modified by a small peptide targeting integrin to selectively boost the tumor-specific oxidative pressure with high efficiency. As shown in Scheme 1, through selective recognition between the peptide and integrin, peptide-modified AuCs enter the cell and then catalyze H_2_O_2_ to produce superoxide anion radicals (O_2_^−●^). It results in the remarkable increase of the ROS level in tumor cells, stimulating mitochondrial apoptosis pathway to release caspases and induce apoptosis.

In a typical experiment, peptide protected AuCs were constructed by a one-step biomimetic mineralization procedure. A peptide with the sequence CCGPDGRDGRDGRDGR was designed as targeting moiety and reducing agent to synthesize AuCs [15]. Obviously, the as-prepared pale-yellow solution of AuCs emits blue fluorescence under 365 nm ultraviolet light radiation (Figure 1a, inset). AuCs exhibits the maximum emission peak localized at 420 nm under a maximum excitation peak at 330 nm, as detected by a fluorescence spectrophotometer (Figure 1a). Moreover, no significant decrease of their fluorescence intensity within 60 days observation (Appendix A), indicating AuCs have a good stability.

Dispersion and size of AuCs were further determined by a high-resolution transmission electron microscopy (HRTEM, Figure 1b) [16]. Since the size of the clusters is considerably ultra-small, they might form larger-sized nanoparticles when exposed under electron beams [17]. The corresponding statistical analysis reveals these nanoparticles are well dispersed with an approximate diameter of 2.85 ± 0.55 nm (Figure 1c), indicating the precursors (the clusters) have a rather satisfying monodispersity. Next, mass spectrometry expediently provides the precise composition of AuCs, that is, the number of noble metal atom and ligand. In our case, matrix-assisted laser desorption/ionization time of flight mass spectrometry (MALDI-TOF MS) was used to characterize the accurate formula of AuCs [18,19,20,21,22,23]. As shown in Figure 1d, MS of AuCs has several main peaks between 1550 and 1800 m/z. These m/z peaks are assigned to Au_7_S_x_Na_y_ (x = 6, 8, 10; y = 0, 1, 2). Especially, one main peak position at 1689 m/z is assigned to Au_7_S_10_. Since there are two cysteine residues in one peptide, one Au_7_ cluster core is inferred coated by five peptides. Therefore, the constitution of the AuCs can be assigned as Au_7_Peptide_5_. Serial peaks can be ascribed to the fragmentation of Au_7_S_10_Na_2_, which may be derived from the cutting effect of the high frequency laser ablation.

### 3.2. Catalytic Capability Analysis

To determine whether AuCs catalyses the endogenous H_2_O_2_ into oxygen radicals under a mimic physiological condition, we applied the compound nitroblue tetrazolium (NBT), a specific substance commonly used to evaluate the level of superoxide anion radicals. NBT has the ability to react with O_2_^−●^ and forms formazan, which is insoluble in water but dissolved in dimethyl sulfoxide (DMSO) with maximum absorption at 680 nm [24].Practically we mixed NBT with AuCs in the presence of 100 µM H_2_O_2_, for the purpose to simulate the H_2_O_2_ level in extra-/intracellular tumor microenvironment. Then, after the formazan deposition was dissolved in DMSO, we explored its optical density (OD) at 680 nm. As indicated in Figure 2a, the OD value increases as the increasing concentration of the catalyst AuCs as expected. Based on these findings, we can draw a conclusion that AuCs are capable of catalyzing the conversion of physiologically levelled H_2_O_2_ into O_2_^−●^, and promising elevate intracellular ROS level.

The catalytic activity of AuCs is affected by pH and temperature. Amplex UltraRed (AUR) assay was definitely used to explore the catalytic activity of AuCs. As depicted in Figure 2b,c the optimal catalytic pH condition is at about pH 7.0, and the optimal temperature is around 40 °C, which are both similar to the physiological environment. Next, under the optimal catalytic conditions, the kinetic parameters were confirmed by Michaelis-Menten and Lineweaver-Burk curves [25] (Figure 2d,e) by changing certain concentrations of AUR and H_2_O_2_, respectively. The result is listed in Table 1, when the substrate is H_2_O_2_, the apparent Michaelis-Menten constant (K_m_) of AuCs isslightly higher than that of natural horseradish peroxidase (HRP) [25], which indicates that AuCs has similar affinity to H_2_O_2_ as HRP. As another comparison, the catalytic constant (K_cat_) of AuCs ismuch smaller than peptide protected gold nanoparticles, which has been previously reported as a peroxidase-mimic enzyme [26]. It may be due to the much enhanced activation site numbers on gold nanoparticles. We also evaluated the kinetic constants by employing AUR as the substrate and further verify the enzyme-like property of AuCs.

### 3.3. Exploring Cytotoxicity Difference

Prior to exploring the cellular toxicity of the clusters and the cytotoxicological mechanism, we investigated the intracellular localization of AuCs on two tumor cells lines. Human primary glioblastoma cell line (U87-MG cells) is expressed with high level of intracellular H_2_O_2_ and high expression of integrin α_ν_β_3_, while human cervical cancer cell line (HeLa cells) was expressed with low level of intracellular H_2_O_2_ and low expression of integrin α_ν_β_3_ [27,28]. The AuCs are co-cultured with above two mentioned cells lines for 12 h and then observed by confocal laser scanning microscopy (CLSM). The CLSM images reveal that clusters are enveloped in U87-MG and HeLa cells, excited to emit blue fluorescence (Appendix A). Then, lysosomal probe was employed to further illustrate the subcellular location of AuCs (Figure 3a). These results suggest that most clusters are located in cytoplasm.

Cellular toxicity of AuCs and peptide were next studied via CCK-8 assay. As shown in Figure 3b,c, the peptide alone doesn’t exert cellular toxicity on two cell lines, whereas AuCs behave obvious cellular toxicity at a dose-dependent manner, when AuCs were incubated with tumor cells for 48 h. Moreover, AuCs’ cellular toxicity on U87-MG cells is significantly higher than that on HeLa cells at the same delivered concentration. Quantatively, the half maximal inhibitory concentration (IC_50_) value of AuCs is around 100 μM for HeLa cells, while that on U87-MG cells is only 40 μM when cells were treated with AuCs for 48 h at 37 °C.

In further tests, two cell lines were incubated with 40 μM AuCs for 24 h and 48 h respectively, and AuCs uptaken by tumor cells was determined by ICP-MS (Figure 4a). The result indicates that the intracellular Au concentration of U87-MG cells is higher than that in HeLa cells at the same incubation time point. It should be noted that, the intracellular Au concentration of U87-MG cells at 24 h incubation is very similar to that incubated with HeLa cells for 48 h. We further compared the relative cytotoxicity of two tumor cells lines at indicated two treatment time points. The cellular toxicity of U87-MG treated with AuCs for 24 h is still more significant higher than that of HeLa cells treated for 48 h (Figure 4b). Since the intrinsic H_2_O_2_ level of U87-MG is higher than that of HeLa cells, it can be inferred that the higher level of ROS isgenerated from AuCs-catalyzed intracellular higher level of H_2_O_2_, and causes much more significant cell toxicity.

### 3.4. Intracellular Biocatalysis

We further detected the change of AuCs-induced intracellular ROS level by using CLSM. As shown in Figure 5, obvious green fluorescence is observed in AuCs treated cells stained by 5-(and-6)-chloromethyl-2′, 7′-dichlorodihydrofluorescein diacetate acetyl ester (CM-H_2_DCFDA), compared with the control. Furthermore, the fluorescence intensity of cells enhances along with the incubation time, indicating significant increase of ROS levels in tumor cells. Then, we further studied the mitochondrial membrane potential. After incubated with AuCs, U87-MG cells were stained by JC-1. Clearly, a significant red to green fluorescence emission variation is observed in most cells (Figure 6a), and the relative fluorescence intensity statistics results are shown in Appendix A. This finding reveals mitochondria depolarization occurs. To illustrate whether the intracellular catalysis could finally induce U87-MG cells apoptosis, we incubated the cells with 20, 40, 60 μM AuCs for 48 h, respectively. Afterwards, an Annexin V-FITC/PI assay was conducted to determine the cell apoptosis ratio. As a result, the apoptosis ratio of tumor cells treated with AuCs is 19.1%, 32.6%, 42.4% (Figure 6b), respectively. The result indicates that AuCs trigger cell apoptosis through a dose-dependent manner. It should be noted that all of them are significantly higher than that in the control group (12.3%). Collectively, the ROS and JC-1 assays mentioned above revealed that AuCs may regulate cell apoptosis through mitochondria-dependent pathway.

### 3.5. Detection Apoptotic Protein Expression

Further, we explored the expression of certain proteins that affect cell apoptosis through a western blot assay (Figure 6c). In which case the expression of caspase-3, caspase-7 and poly (ADP-ribose) polymerase (PARP) do not show any increasing tendency, whereas the expression of their active forms rises by AuCs, dose dependently. Hence, we can draw a conclusion that the clusters triggers cell apoptosis through a mitochondria-dependent pathway, through which the downstream protein caspase-3, caspase-7 and PARP were cut into their active forms. Collectively, the catalytic generated ROS is prone to depolarize mitochondria and activates caspase protein, which in turn causes tumor cell apoptosis [29].

## 4. Conclusions

In summary, we have demonstrated peptide-coated AuCs, as a novel intracellular catalyst, convert low-toxic endogenous H_2_O_2_ into higher-toxic O_2_^−●^ under mild physiological condition. The steady-state kinetic studies show that AuCs promotes the catalytic reaction at the mimic tumor microenvironment in an efficient manner. Importantly, compared to HeLa cells, due to ligands with selective recognition, these AuCs were preferably internalized by U87-MG cells overexpressing relatively higher levelled integrin and H_2_O_2_. Correspondingly, more ROS with higher activity were produced from endogenous H_2_O_2_ to induce mitochondria-dependent apoptosis. As a result, higher efficiency of anticancer therapy was achieved. These findings indicate such AuCs hold a great promise for anticancer treatment in vitro. This study will open a new venue to explore metal nanoclusters for cancer therapy through catalytic approaches.

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
