# Peer review of "Peptide-Templated Gold Clusters as Enzyme-Like Catalyst Boost Intracellular Oxidative Pressure and Induce Tumor-Specific Cell Apoptosis"

_nanomaterials, 2018, doi:10.3390/nano8121040_

Round 1
Reviewer 1 Report
Authors reported target-specific apoptosis using gold clusters templated with peptides, which catalyze superoxide anion formation that results in a high level of reactive oxygen species, a lower mitochondrial membrane potential, and finally cancer cell death. Their conclusions are well supported by cellular toxicity and intracellular catalysis experiments. The anticancer therapy is very interesting and in-vivo tests are strongly recommended. The comments are the following.
1. Au cluster size
TEM imaging of gold clusters, more or less than 1 nm, is challenging as mentioned, but the Ref. [17] couldn't be a literature to support what they observed in TEM (Figure 1b). Only the structural change was reported, neither aggregation nor bigger particles. Instead, correlating the results about fluorescence color and spectra or MS analysis may be better. Actually it has been well known that nano clusters could be stable when encapsulated in such as capping polymers. Their Au Cs are also protected by peptides, but seem not to be enough for stability under the electron beam. SEM on STEM analysis is worth to try.
2. Catalytic activity of AuCs
Amplex UltraRed (AUR) reacts with H2O2 in 1:1 stoichiometric ratio to produce Amplex UltroxRed with emission peak at 581, not 680 nm. They might confuse it with formazan on page 7. In addition, Amplex UltraRed already has dependence on pH. Hence the optimal catalytic temperature and pH they observed might be due to AUR rather than Au Cs. In order to see how much Au Cs catalyze the oxidation of AUR, such a blank test is required. For example, Figures 2d and 2e experiments without AuCs.
3. Typo "Casepase-3" in Figure 6c
Author Response
Response to the Reviewer 1:
Q1. Au cluster size
TEM imaging of gold clusters, more or less than 1 nm, is challenging as mentioned, but the Ref. [17] couldn't be a literature to support what they observed in TEM (Figure 1b). Only the structural change was reported, neither aggregation nor bigger particles. Instead, correlating the results about fluorescence color and spectra or MS analysis may be better. Actually it has been well known that nano clusters could be stable when encapsulated in such as capping polymers. Their Au Cs are also protected by peptides, but seem not to be enough for stability under the electron beam. SEM on STEM analysis is worth to try.
A1. We have changed Ref. [17] into the following one. Huseyinova, S.; Blanco, J.; Requejo, F. G.; Ramallo-López, J. M.; Blanco, M. C.; Buceta, D.; & López-Quintela, M. A. Synthesis of highly stable surfactant-free Cu5 clusters in water. J. Phys. Chem. C 2016, 120, 15902-15908.
In this paper, the authors discussed that the clusters are easily aggregated to form bigger nanopartiles under the irradiation of the electron beam.
The resolution of SEM is definitely worse than HRTEM. The resolution of STEM is similar to HRTEM. Therefore, perhaps HRTEM is the preferred technique to characterize metal clusters.
Q2. Catalytic activity of AuCs
Amplex UltraRed (AUR) reacts with H2O2 in 1:1 stoichiometric ratio to produce Amplex UltroxRed with emission peak at 581, not 680 nm. They might confuse it with formazan on page 7. In addition, Amplex UltraRed already has dependence on pH. Hence the optimal catalytic temperature and pH they observed might be due to AUR rather than Au Cs. In order to see how much Au Cs catalyze the oxidation of AUR, such a blank test is required. For example, Figures 2d and 2e experiments without AuCs.
A2. The maximum fluorescence emission peak of Amplex UltroxRed has been corrected as 585 nm in the revised edition.
We admit the fluorescence of the oxidized product of Amplex UltraRed is dependent on pH. Normally, the oxidizing substrate is hydrogen peroxide. We referred to the product specification of Amplex® UltraRed Reagent from invitrogen and found that Amplex® UltraRed remains stable across a broader pH range, from pH 5.5 to 9.0 (refered to the following Figure). However, in our case, AuCs behave the most significant peroxidase-like activity at around pH 7.0. We suppose this pH-dependent peroxidase-like activity is ascribed to AuCs, rather than due to AUR.
We compared the fluorescence intensity of Amplex UltraRed oxidized with and without AuCs. It is noted that we selected the lowest concentration of H2O2 and Amplex UltraRed in this case. The results show that the fluorescence intensity of the mixture with AuCs is obviously higher than that without AuCs (compared the FL intensity of black bar VS red bar, blue bar VS green bar in the following Figure). It can be inferred that the background interference can be ignored.
Q3. Typo "Casepase-3" in Figure 6c.
A3. We have corrected the spelling of the word.
Response to the Reviewer 2:
In this manuscript, reactive oxygen species is formed from H2O2 via peptide-templated Au clusters as catalyst and is used for apotosis of tumor cells. The peptide can be bound to integrin on the tumor cell. The Au clusters are more effective against U87-MG cell than HeLa cell, because the former has more the integrin and H2O2 than the latter.
The intracellular H2O2 effect is confirmed by the comparison between U87-MG for the incubation time = 24h and HeLa for 48h, in which the concentration of Au were almost the same. The apoptosis proceeds via mitochondria-dependent pathway, which was confirmed by JC-1 assay. The mechanism of the apoptosis using the Au cluster is properly proved. Thus this is publishable in the journal. However, there are small problems as follows.
Q1. In Fig. 2b, the number of the top on the y-axis was shown as “0”.
A1. We have ajusted the position of the Figure 2b. The maximum number of the y-axis is shown as 100.
Q2. “Hela cells” -> “HeLa cells”
A2. We have corrected the spelling.
Q3. In the line 271, the authors should start new paragraph from “In further tests”.
A3. We have started the new paragraph according to the suggestion from the reviewer.
Q4. In Fig 3b/c, why were the cell viability increased by the addition of only peptide?
A4. The reason might be ascribed to that the peptide might be hydrolyzed by protease in cell culture medium. It is propable that the degraded oligopeptides or amino acid have become the nutritive material to foster the tumor cells.
Q5. To simplify the meaning for comparison between U87-MG for 24h and HeLa for 48h, it may be better that the color and pattern In Fig.4a fit those in Fig.4b.
A5. The color and pattern of the bars in two figures have been adjusted same to each other.
Q6. In Fig.6a, the authors should show not only the image, but also the graph of the fluorescence intensities from JC-1 monomer and aggregate at the various time.
A6. We have analyzed the relative fluorescence intensity of cells stained by JC-1 monomer and aggregate. Please refer to Figure S3.
Figure S3. Statistical results of relative fluorescence intensity indicating mitochondrial membrane potential change. The ratio of green and red fluorescence intensity of cells incubated with AuCs for 12 hours was set to 100%. *** P < 0.01.

Reviewer 2 Report
In this manuscript, reactive oxygen species is formed from H2O2 via peptide-templated Au clusters as catalyst and is used for apotosis of tumor cells. The peptide can be bound to integrin on the tumor cell. The Au clusters are more effective against U87-MG cell than HeLa cell, because the former has more the integrin and H2O2 than the latter.
The intracellular H2O2 effect is confirmed by the comparison between U87-MG for the incubation time = 24h and HeLa for 48h, in which the concentration of Au were almost the same. The apoptosis proceeds via mitochondria-dependent pathway, which was confirmed by JC-1 assay. The mechanism of the apoptosis using the Au cluster is properly proved. Thus this is publishable in the journal.
However, there are small problems as follows.
In Fig. 2b, the number of the top on the y-axis was shown as “0”.
“Hela cells” -> “HeLa cells”
In the line 271, the authors should start new paragraph from “In further tests”.
In Fig 3b/c, why were the cell viability increased by the addition of only peptide?
To simplify the meaning for comparison between U87-MG for 24h and HeLa for 48h, it may be better that the color and pattern In Fig.4a fit those in Fig.4b,
In Fig.6a, the authors should show not only the image, but also the graph of the fluorescence intensities from JC-1 monomer and aggregate at the various time.
Author Response
Response to the Reviewer 1:
Q1. Au cluster size
TEM imaging of gold clusters, more or less than 1 nm, is challenging as mentioned, but the Ref. [17] couldn't be a literature to support what they observed in TEM (Figure 1b). Only the structural change was reported, neither aggregation nor bigger particles. Instead, correlating the results about fluorescence color and spectra or MS analysis may be better. Actually it has been well known that nano clusters could be stable when encapsulated in such as capping polymers. Their Au Cs are also protected by peptides, but seem not to be enough for stability under the electron beam. SEM on STEM analysis is worth to try.
A1. We have changed Ref. [17] into the following one. Huseyinova, S.; Blanco, J.; Requejo, F. G.; Ramallo-López, J. M.; Blanco, M. C.; Buceta, D.; & López-Quintela, M. A. Synthesis of highly stable surfactant-free Cu5 clusters in water. J. Phys. Chem. C 2016, 120, 15902-15908.
In this paper, the authors discussed that the clusters are easily aggregated to form bigger nanopartiles under the irradiation of the electron beam.
The resolution of SEM is definitely worse than HRTEM. The resolution of STEM is similar to HRTEM. Therefore, perhaps HRTEM is the preferred technique to characterize metal clusters.
Q2. Catalytic activity of AuCs
Amplex UltraRed (AUR) reacts with H2O2 in 1:1 stoichiometric ratio to produce Amplex UltroxRed with emission peak at 581, not 680 nm. They might confuse it with formazan on page 7. In addition, Amplex UltraRed already has dependence on pH. Hence the optimal catalytic temperature and pH they observed might be due to AUR rather than Au Cs. In order to see how much Au Cs catalyze the oxidation of AUR, such a blank test is required. For example, Figures 2d and 2e experiments without AuCs.
A2. The maximum fluorescence emission peak of Amplex UltroxRed has been corrected as 585 nm in the revised edition.
We admit the fluorescence of the oxidized product of Amplex UltraRed is dependent on pH. Normally, the oxidizing substrate is hydrogen peroxide. We referred to the product specification of Amplex® UltraRed Reagent from invitrogen and found that Amplex® UltraRed remains stable across a broader pH range, from pH 5.5 to 9.0 (refered to the following Figure). However, in our case, AuCs behave the most significant peroxidase-like activity at around pH 7.0. We suppose this pH-dependent peroxidase-like activity is ascribed to AuCs, rather than due to AUR.
We compared the fluorescence intensity of Amplex UltraRed oxidized with and without AuCs. It is noted that we selected the lowest concentration of H2O2 and Amplex UltraRed in this case. The results show that the fluorescence intensity of the mixture with AuCs is obviously higher than that without AuCs (compared the FL intensity of black bar VS red bar, blue bar VS green bar in the following Figure). It can be inferred that the background interference can be ignored.
Q3. Typo "Casepase-3" in Figure 6c.
A3. We have corrected the spelling of the word.
Response to the Reviewer 2:
In this manuscript, reactive oxygen species is formed from H2O2 via peptide-templated Au clusters as catalyst and is used for apotosis of tumor cells. The peptide can be bound to integrin on the tumor cell. The Au clusters are more effective against U87-MG cell than HeLa cell, because the former has more the integrin and H2O2 than the latter.
The intracellular H2O2 effect is confirmed by the comparison between U87-MG for the incubation time = 24h and HeLa for 48h, in which the concentration of Au were almost the same. The apoptosis proceeds via mitochondria-dependent pathway, which was confirmed by JC-1 assay. The mechanism of the apoptosis using the Au cluster is properly proved. Thus this is publishable in the journal. However, there are small problems as follows.
Q1. In Fig. 2b, the number of the top on the y-axis was shown as “0”.
A1. We have ajusted the position of the Figure 2b. The maximum number of the y-axis is shown as 100.
Q2. “Hela cells” -> “HeLa cells”
A2. We have corrected the spelling.
Q3. In the line 271, the authors should start new paragraph from “In further tests”.
A3. We have started the new paragraph according to the suggestion from the reviewer.
Q4.In Fig 3b/c, why were the cell viability increased by the addition of only peptide?
A4. The reason might be ascribed to that the peptide might be hydrolyzed by protease in cell culture medium. It is propable that the degraded oligopeptides or amino acid have become the nutritive material to foster the tumor cells.
Q5. To simplify the meaning for comparison between U87-MG for 24h and HeLa for 48h, it may be better that the color and pattern In Fig.4a fit those in Fig.4b.
A5. The color and pattern of the bars in two figures have been adjusted same to each other.
